# Study on Mix Proportion Optimization of Manufactured Sand RPC and Design Method of Steel Fiber Content under Different Curing Methods

**DOI:** 10.3390/ma12111845

**Published:** 2019-06-06

**Authors:** Chunling Zhong, Mo Liu, Yunlong Zhang, Jing Wang

**Affiliations:** 1School of Economics and Management, Jilin Jianzhu University, Changchun 130000, China; zhongchunling@jlju.edu.cn; 2School of Civil Engineering, Jilin Jianzhu University, Changchun 130000, China; lium547573@163.com; 3School of Transportation Science and Engineering, Jilin Jianzhu University, Changchun 130000, China

**Keywords:** manufactured sand, compressive strength, splitting tensile strength, steel fiber contents

## Abstract

This study investigated four factors (water/binder ratio, silica fume, fly ash, and sand/binder ratio) using the orthogonal experimental design method to prepare the mix proportions of a manufactured sand reactive powder concrete (RPC) matrix to determine the optimal matrix mix proportions. On this basis, we assessed the compressive and splitting tensile strengths of different steel fiber contents under natural, standard, and compound curing conditions to develop an economical and reasonable RPC for various engineering requirements. A calculation method for the RPC strength of the steel fiber contents was evaluated. The results showed that the optimum steel fiber content for manufactured sand RPC is 4% under natural, standard, and compound curing conditions. Compared with standard curing, compound curing can improve the early strength of manufactured sand RPC but only has a small effect on the enhancement of late strength. Although the strength of natural curing is slightly lower than that of standard curing, it basically meets project requirements and is beneficial for practical applications. The calculation formula of 28-day compressive and splitting tensile strengths of manufactured sand RPC steel fiber at 0%–4% is proposed to meet the different engineering requirements and the flexible selection of steel fiber content.

## 1. Introduction

Reactive powder concrete (RPC) is a new cement-based composite material developed by Bouygues in France in the 1990s. RPC is produced by eliminating coarse aggregate (to optimize the compactness of quartz sand aggregate); mixing with silica fume, fly ash, and mineral powder (to reduce porosity); using short fine steel fiber (to improve material ductility); and heat curing (to improve the material’s microstructure). This procedure produces cement-based materials of high strength, high durability, high toughness, and good volume stability [1,2,3]. Existing studies have used quartz sand [4,5,6] and river sand [7,8,9] to prepare RPC. However, quartz sand increases preparation costs, and the massive utilization of river sand causes shortages in resources. Manufactured sand is an environmentally friendly material advocated by China. Researchers have evaluated the use of manufactured sand to prepare high-performance [10], self-compacting [11], and lightweight concrete [12]. However, no research has been reported on the preparation of RPC using manufactured sand. Therefore, the mix proportions and mechanical properties of RPC should be investigated using manufactured sand.

Existing studies have indicated that curing methods have an important effect on the mechanical properties of RPC. No standard RPC curing conditions have been reported. In general, two kinds of curing are used for RPC preparation, namely, heat curing and autoclave curing. Heat curing includes steam curing, hot water curing, and high-temperature curing [13]. This method can effectively change the RPC microstructure and improve its mechanical properties through RPC hydration and enhancing the activity of pozzolan [14]. At the same time, heating time and rate are important factors used in thermal curing, which can affect the crystallization of hydrates, pozzolanic reaction, and microstructure formation [15]. Autoclave curing is conducted by simultaneously applying heat and pressure on fresh samples. Different preset pressures have a certain influence on RPC compressive strength. The compressive strength of autoclave curing can reach 500 MPa by applying 60 MPa pre-pressure and 250 °C heat treatment [2]. However, high pre-pressure does not indicate considerable compressive strength. Ipek et al. systematically investigated the influence of preset pressure on the mechanical properties of RPC. The results showed that the compressive strength reached 475 MPa when the pre-pressure was 100 MPa. In addition, they showed the adverse effect of high presetting pressure at 125 MPa. The reduced strength was due to the huge expansion of microcracks around the aggregates after the release of pressure at 125 MPa [16]. Heat curing and autoclave curing are unsuitable for on-site construction because they are limited to laboratory and factory prefabrication, which limits the promotion and application of RPC. Therefore, manufactured sand RPC prepared under natural, standard, and compound curing conditions should be investigated to provide a basis for the application of RPC in practical engineering.

The mechanical properties of RPC are remarkably improved, its failure mode is changed, and its ductility is significantly improved due to the incorporation of steel fibers [17]. Researchers have conducted numerous studies on the effect of steel fiber on mechanical properties. Studies have shown that steel fiber content and type of compressive strength have obvious enhancement effects [18]. However, the conclusions differ due to variations in the size and type of steel fiber used [19,20]. The compressive strength of RPC is slightly improved when the steel fiber content increases from 0% to 3% [21]. However, the compressive strength of RPC decreases and the tensile strength slightly increases when the steel fiber content exceeds 3.5% [22]. In addition, different types of steel fiber have varied effects on the mechanical properties of RPC. The compressive strength can reach 150 MPa or more when 3% ordinary steel fiber is blended, and the 28-day compressive strength of RPC with 3% end hook and corrugated fiber is increased by 48% and 59%, respectively [9]. Huang et al. showed that improving the orientation of steel fiber could significantly improve the mechanical properties. Based on experimental results, linear relationships were regressed for fiber orientation parameters and mechanical properties [23]. Previous studies have focused on the influence of the type, content, and orientation of steel fiber on the mechanical properties of RPC. Existing research results have shown that steel fiber content has a large influence on the strength of RPC. However, whether steel fiber content can be designed based on actual project requirements has not been reported. Therefore, we conducted an experiment on the RPC compressive and splitting tensile strengths of different steel fiber contents and devised a formula that can flexibly select the steel fiber content based on the actual requirements.

In summary, based on the orthogonal experimental design (OED) method, this study used the water/binder ratio, silica fume, fly ash, and sand/binder ratio as factors to prepare a manufactured sand RPC matrix and determined the optimal matrix mix proportions. The mechanical properties of the manufactured sand RPC and the design method of the steel fiber content were studied by adding different steel fiber contents. The 7- and 28-day compressive and splitting tensile strengths of five steel fiber contents under natural, standard, and compound curing conditions were evaluated by laboratory tests, and the microscopic mechanism was analyzed by SEM. On this basis, the relationships between steel fiber content and compressive strength and between steel fiber content and splitting tensile strength are summarized here and can be used to provide a reliable steel fiber content scheme for the design and construction of RPC structures.

## 2. Materials and Methods

### 2.1. Raw Materials

This study adopted P.II-grade 52.5 cement produced by Yatai Cement Industry (Changchun, China). The cement inspection report, which conforms to the general Portland Cement Inspection Standard (GB175-2007) [24], is shown in Table 1. Silica fume produced by Dongyue Silicone Material (Zibo, China), fly ash produced by Datang Second Thermal Power (Changchun, China), and S95-grade mineral powder produced by Longze Water Purification Materials (Gongyi, China) were used. The test reports are shown in Table 2, Table 3 and Table 4. All of these were in accordance with the technical specifications of the Application of Mineral Admixtures (GB/T51003/2014) [25]. The superplasticizer used was HSC polycarboxylic acid high-performance water reducer produced by Hongxia Polymer Materials (Qingdao, China). The inspection indexes are shown in Table 5. The fine aggregate was made of manufactured sand produced by Jiusheng Industry (Changchun, China), a picture of which is shown in Figure 1. The screening results are shown in Table 6, and according to the Standard for Technical Requirements and Test Method of Sand and Crushed Stone for Ordinary Concrete (JGJ 52-2006) [26], the draw gradation curve is shown in Figure 2. The steel fiber was made of copper-plated steel fiber produced by Zhitai Steel Fiber Industry (Tangshan, China). The specifications and performance indexes of the fine aggregate are shown in Table 7, and a picture is shown in Figure 3.

### 2.2. Sample Preparation

The mixing of RPC is different from the conventional concrete mixing process. The specific preparation process is shown in Figure 4.

The test used indoor natural curing (NC) (15 ± 2 °C), standard curing (SC) (20 ± 2 °C, 95% humidity), and compound curing (CC), as shown in Figure 5, and their specific processes are shown in Table 8.

### 2.3. Testing Procedure

#### 2.3.1. Orthogonal Experimental Design

Orthogonal experimental design is a test method suitable for multifactor and multilevel comparison tests. As shown in Figure 6, it selects some representative sample from the comprehensive test according to orthogonality to have uniform and comparable dispersion. The characteristics of the test cannot fully consider the impact of many factors on the performance indicators in the test, can reduce the number of trials to a greater extent, and can facilitate scientific judgments and analyses of complex test results.

This study used a four-factor and three-level orthogonal test. As shown in Table 9, the water/binder ratio, silica fume content, fly ash content, and sand/binder ratio were taken as four factors, and three levels were set for each factor.

#### 2.3.2. Mechanical Properties Tests

Specimens (100 × 100 × 100 mm) were selected as the test piece based on GB/T31387-2015 [27] to evaluate the compressive and splitting tensile strengths. The samples were tested by SYE-3000B (hydraulic press of new testing machine Co. Ltd, Changchun, China), for 7 and 28 days, and the average of the three measurements was used as the compressive strength and splitting tensile strength of the test specimens. The compressive test loading rate was maintained between 1.2 and 1.4 MPa/s, and the loading rate of the splitting tensile strength test was maintained between 0.08 and 0.1 MPa/s.

#### 2.3.3. Microstructural Tests

Samples for SEM were small pieces (about 5 mm) taken from the compressive failure specimens. Through drying, vacuum pumping, and gold spraying, the RPC microstructures were observed and photographed using a scanning electron microscope TM3030 of Hitachi (Tokyo, Japan).

## 3. Results and Discussion

### 3.1. Design and Analysis of Mixing Proportions of Manufactured Sand RPC Matrix

#### 3.1.1. Experimental Design and Test Results Based on OED

The effects of the water/binder ratio, silica fume content, fly ash content, and sand/binder ratio on the 28-day compressive strength of the manufactured sand RPC matrix were studied by OED. The experimental design and results are shown in Table 10.

#### 3.1.2. Statistical Analysis and Discussion

The test results were analyzed by the range analysis method, and the range effect values of the water/binder ratio, silica fume content, fly ash content, and sand/binder ratio on the 28-day compressive strength of the manufactured sand RPC matrix were analyzed. The results are shown in Table 11.

The range value R indicates the influence of this factor on the test index. It can be seen from the range analysis of compressive strength in Table 12 that the primary and secondary relationships of the influence of four factors on the 28-day compressive strength of the RPC matrix under NC, SC, and CC were all B > A > D > C (i.e., silica fume content > water/binder ratio > sand/binder ratio > fly ash content). Mixing proportions of manufactured sand RPC matrix were A_2_B_2_C_3_D_1_ (i.e., water/binder ratio: 0.18, silica fume content: 13%, fly ash content: 20%, and sand/binder ratio: 0.7).

As shown in Figure 7a, as the water/binder ratio decreased, the 28-day compressive strength of the manufactured sand RPC matrix first increased and then decreased. This is because under the conditions of compaction, the lower the water/binder ratio, the higher the compressive strength of the matrix. However, when the water/binder ratio was 0.16, because the water/binder ratio was too low, there was not enough water to fully hydrate the cement, resulting in a decrease in the compressive strength of the matrix.

As shown in Figure 7b, with the increase of the silica fume content, the 28-day compressive strength of the manufactured sand RPC matrix first increased and then decreased. This is because of the microaggregate effect of silica fume. Compared with cement, silica fume is thinner and can be fully filled into the cement gap, which greatly promotes the density of the cement matrix and increases its compressive strength.

As shown in Figure 7c, with the increase of the fly ash content, the 28-day compressive strength of the manufactured sand RPC matrix gradually increased. This is because the particle size of fly ash is small, and the surface is smooth and spherical. During mixing, fly ash produces a “ball effect”, which plays a role in filling the particle gap and improving the density of the slurry, thus improving the compressive strength of the matrix.

As shown in Figure 7d, the 28-day compressive strength of the manufactured sand RPC matrix gradually decreased with the increase of the sand/binder ratio. The effect of the sand/binder ratio on the compressive strength of the matrix was essentially the effect of the average slurry thickness on the compressive strength of the matrix. When the sand/binder ratio decreased, the average slurry thickness increased, so that there was enough slurry between the sand grains to connect them as a whole. Also, the bonding force between the sand grains and the slurry increased, so that the compressive strength of the matrix increased. On the contrary, the compressive strength of the matrix decreased with the increase of the sand/binder ratio and the decrease of average slurry thickness.

### 3.2. Study on Mechanical Properties of Manufactured Sand RPC

#### 3.2.1. Experiment Design

On the basis of the optimal matrix mix proportions of manufactured sand RPC, steel fibers with volume contents of 0%, 1%, 2%, 3%, 4%, and 5% were added. The mechanical properties of manufactured sand RPC with different steel fiber contents under different curing methods were further studied.

#### 3.2.2. Manufactured Sand RPC Failure Mode

The compression test shows that the undoped steel fiber specimens were accompanied by a large cracking sound during breakage and a large amount of peeling. The specimens were deformed in a conical shape and were in brittle failure mode. The steel fibers had a restraining effect on the transverse direction of the test block when they were incorporated into manufactured sand RPC. At the initial stage of the test block, the tensile stress of the steel fibers was small due to the small strain occurring with the increase in the load, and the concrete matrix played a major role in the force. In the last stage, the steel fibers that crossed the crack began to work with the increase in the concrete strain by enduring most of the stresses and alleviating the degree of stress concentration, thereby enabling the material to bear the cracks. No large-scale peeling occurred, only a small amount of debris fall and cracks with different degrees appeared, and good integrity was maintained when the test block reached the maximum load.

The undoped steel fiber specimens were longitudinally split and fractured. The test piece was no longer brittle and cracked with the incorporation of steel fibers. The mechanisms of the influence of steel fibers on failure mode were toughening and cracking effects. In the splitting tensile failure test, the crack section was basically a flat straight line, and the crack first appeared in the matrix with weak tensile strength. The matrix and steel fibers were subjected to tensile stress due to the presence of microcracks. The width of the microcracks gradually increased with the increase in stress. Tensile stress was transmitted to the concrete on the two sides of the crack due to the bonding stress of the steel fibers and the matrix. Bonding failure occurred when the tensile stress exceeded the bonding stress of the steel fibers to the substrate.

#### 3.2.3. Test Results of the Mechanical Properties of Manufactured Sand RPC

The test results of compressive and splitting tensile strengths of different steel fiber contents under natural, standard, and compound curing of manufactured sand RPC for 7 and 28 days are shown in Table 12.

#### 3.2.4. Analysis of the Mechanical Properties of Manufactured Sand RPC

The test results of compressive and splitting tensile strengths of manufactured sand RPC are shown in Figure 8 and Figure 9.

Figure 8 and Figure 9 show the 7- and 28-day compressive and splitting tensile strengths of manufactured sand RPC under natural, standard, and compound curing conditions. The effect of the three curing conditions on 7-day compressive and splitting tensile strengths was CC > SC > NC, and the effect of 28-day compressive and splitting tensile strengths was SC > CC > NC.

The effects of three different curing methods on the mechanical properties of manufactured sand RPC were based on standard curing. Figure 10 shows the proportion of compressive and splitting tensile strengths under natural and standard curing at 7 and 28 days and the proportion of compressive and splitting tensile strengths under composite curing and standard curing at 7 and 28 days. Under 7 days of natural curing, the compressive and splitting tensile strengths of manufactured sand RPC exceeded 90% of standard curing. The ratio of 28-day natural curing to standard curing compressive strength was more than 90%, and the ratio of tensile splitting strength was more than 80%. The compressive and splitting tensile strengths of manufactured sand RPC accounted for more than 110% and 90% of standard curing under 7- and 28-day compound curing, respectively. These findings were due to the high percentage of hydration products produced through rapid hydration of compound curing in the previous 1 day of hot water curing. This high percentage promoted the development of hydration products and accelerated the pozzolanic reaction of silica fume and Ca(OH)_2_, which caused 7-day compound curing to have enhanced strength. However, in the later stages of natural curing, the hydration and formation of crystalline products of manufactured sand RPC were hindered by water evaporation. Therefore, the strength of 28-day compound curing became slightly lower than that of standard curing, and the ratio of 7-day compound curing to standard curing strength was higher than that of the 28-day strength ratio.

The influence of different steel fiber contents on the compressive and splitting tensile strengths of manufactured sand RPC was different. Figure 11 and Figure 12 show the percentage improvement of the compressive and splitting tensile strengths of manufactured sand RPC based on steel fiber content. The compressive and splitting tensile strengths of the three curing methods at 7 and 28 days first increased and decreased with the increase of steel fiber content. The compressive and splitting tensile strengths gradually increased when the steel fiber content was 0%–4%, whereas the compressive and splitting tensile strengths decreased when the steel fiber content was 5%. The percentage increase in compressive and splitting tensile strengths with the increase in steel fiber content was analyzed by using undoped steel fiber as a reference. The results showed that steel fibers did not significantly improve the compressive strength of manufactured sand RPC. At 7 days, the increase in compressive strength was the smallest under natural curing (4%–10%) and the largest under compound curing (6%–27%). At 28 days, the increase in compressive strength was the smallest under compound curing (10%–24%) and the largest under standard curing (14%–31%). However, the steel fibers remarkably increased the splitting tensile strength of manufactured sand RPC. At 7 days, the increased rate of splitting tensile strength under natural curing was the smallest (38%–170%), and the increased rate of splitting tensile strength under compound curing was the largest (42%–190%). At 28 days, the increased rate of splitting tensile strength under compound curing was the smallest (42%–130%), and the increased rate of splitting tensile strength under standard curing was the largest (56%–175%). Steel fibers improved the strength of manufactured sand RPC because their strength was higher than that of the RPC matrix and they were larger than the tip of the crack. Thus, steel fibers played bridging and pinning roles to hinder the development of cracks. Therefore, an appropriate amount of steel fiber can increase the strength of manufactured sand RPC to a certain extent. However, steel fibers tend to agglomerate and cannot be completely utilized when their content is high, resulting in the decrease in the strength of manufactured sand RPC.

#### 3.2.5. Microscopic Mechanism Analysis

The macroscopic mechanical properties of matter are related to composition and microstructure. Thus, the analysis of the macroscopic properties of manufactured sand RPC should be combined with the microstructure formation. On the basis of the SEM analysis of manufactured sand RPC, the RPC matrix of manufactured sand had no obvious pores and defects, the porosity was low, and the hydration products were mainly C–S–H gel, as shown in Figure 13a. The interface transition zone between steel fibers and the matrix was close, as shown in Figure 13b.

As shown in the SEM image in Figure 13b, the surface of the steel fibers became smooth due to copper plating, and only a small amount of hydration products were attached to the surface of the fibers. Moreover, the steel fibers and the surrounding matrix had a firm structure, without an interfacial transition zone and good compactness. Inside the RPC matrix, the hydrated calcium silicate gel was easily deposited and tightly bonded on the surface of steel fibers. The steel fibers had a high elastic modulus when RPC was under external load, and the adhesion between the RPC matrix had a certain role in preventing crack development. Therefore, steel fibers played bridging and pinning roles to reduce the microcracks at the interface, which improved the interface strength of RPC and the crack resistance of concrete.

### 3.3. Steel Fiber Content Design

#### 3.3.1. Compressive Strength Design

To determine the relationship between steel fiber content and compressive strength, the percentage of steel fiber blending in the three curing methods was averaged based on the increase of compressive strength of manufactured sand RPC at 28 days and fitted, as shown in A of Figure 14. When the content of steel fiber was 5%, the preparation cost increased and the compressive strength decreased. It is suggested that the content of steel fiber should be between 0% and 4% in practical engineering. Because the contribution rate of 0%–4% steel fiber content to the compressive strength of manufactured sand RPC was close to a straight line, in order to facilitate the engineering design, linear fitting was adopted in this paper, as shown in B of Figure 14. The formula for calculating the compressive strength of manufactured sand RPC at 28 days was obtained based on the percentage increase of compressive strength:(1)$$f=\left(1+\lambda_{c}\right)f_{0}$$
where *f* is 28-day compressive strength (MPa), $\lambda_{c}$ is the steel fiber compressive strength contribution, $\lambda_{c}=\frac{6.2x+3.15}{100}$, *f*_0_ corresponds to the 28-day compressive strength of the undoped steel fiber under curing mode (MPa), and x is the steel fiber content (%).

The experimental results in this study were compared with the calculated results to verify the applicability of the manufactured sand RPC compressive strength formula, as shown in Table 13.

To verify the feasibility of the relationship between steel fiber content and compressive strength, Guo [28] investigated the effect of steel fiber content on RPC strength under different curing conditions. The calculated values of the proposed formula were compared with their test results, as shown in Table 14.

Ma [17] used river sand as a fine aggregate to determine the compressive strength of different steel fiber blends after standard curing for 28 days. The calculated values of the proposed formula were compared with their test results, as shown in Table 15.

As shown in Table 13 and Table 14, the maximum error between the calculated and actual values was 6.7%, indicating that the degree of compliance with the test results was good. Therefore, the proposed formula can be applied to the calculation of 28-day compressive strength when the steel fiber content in RPC is 0%–4% under natural, standard, hot water, and steam curing conditions using manufactured sand and quartz sand as the fine aggregate. As shown in Table 15, the maximum difference between the calculated and actual values was 18.55% but less than 20%. The proposed formula can serve as a guide, although it has a large error in predicting the compressive strength of RPC prepared with river sand.

#### 3.3.2. Splitting Tensile Strength Design

To determine the relationship between steel fiber content and splitting tensile strength, the percentage of steel fiber blending in the three curing methods was averaged based on the increase of cracking tensile strength of manufactured sand RPC at 28 days and fitted, as shown in A of Figure 15. When the content of steel fiber was 5%, the preparation cost increased and the splitting tensile strength decreased. It is suggested that the content of steel fiber should be between 0% and 4% in practical engineering. Because the contribution rate of 0%–4% steel fiber content to the splitting tensile strength of manufactured sand RPC was close to a straight line, in order to facilitate the engineering design, linear fitting was adopted in this paper, as shown in B of Figure 15. The formula for calculating the tensile strength of manufactured sand RPC at 28 days was obtained based on the percentage fitting formula of splitting tensile strength increase:(2)$$f=\left(1+\lambda_{t}\right)f_{0}$$
where *f* is 28-day splitting tensile strength (MPa), $\lambda_{t}$ is the steel fiber splitting tensile strength contribution, $\lambda_{t}=\frac{35.44x+6.92}{100}$, *f*_0_ corresponds to the 28-day splitting tensile strength of the undoped steel fiber under curing mode (MPa), and x is the steel fiber content (%).

The applicability of the cracking tensile strength formula of manufactured sand RPC was verified. Only the experimental data were used to verify the proposed tensile strength formula because the tensile strength of RPC was immensely affected by the size and type of steel fibers. The calculated values were compared with the experimental values, as shown in Table 16.

As shown in Table 16, the error between the calculated and actual values was within 10%, and the degree of compliance with the test results was good. The proposed formula is suitable for the calculation of 28-day splitting tensile strength of mechanical fiber with fine-grained aggregate as the fine aggregate under natural, standard, and compound curing conditions with 0%–4% steel fiber content.

## 4. Conclusions

In this study, manufactured sand was used as the fine aggregate to prepare RPC under natural, standard, and compound curing conditions. The mechanical properties of manufactured sand and the design of the steel fiber content were investigated through the evaluation of compressive and split tensile strengths. The conclusions are summarized as follows.

Based on the orthogonal experimental design, the optimum mixing proportions of manufactured sand RPC matrix were water/binder ratio: 0.18, silica fume content: 13%, fly ash content: 20%, and sand/binder ratio: 0.7.The compressive and splitting tensile strengths of manufactured sand RPC increased to a certain extent, the impact of the increase of compressive strength was small, and the increase in splitting tensile strength was significant when the steel fiber content was 0%–4%. SEM analysis showed that the porosity of the manufactured sand RPC was extremely low, the hydration products were mostly C–S–H gel, the interface transition zone between steel fibers and the matrix was close.Compared with standard curing, compound curing improved the early strength of manufactured sand RPC and did not improve the later strength. Natural curing basically meets the engineering requirements and is beneficial for practical applications, although its strength is slightly lower than standard curing. The percentage of compressive strength and splitting tensile strength increase of manufactured sand RPC 7 days under three curing methods was CC > SC > NC, and the order of increase of compressive strength and splitting tensile strength for 28 days was SC > NC > CC.Prediction formulas of 28-day compressive and splitting tensile strengths of steel fibers with 0%–4% were established to aid the selection of steel fiber content based on different actual demands.

## Figures and Tables

**Figure 1 materials-12-01845-f001:**
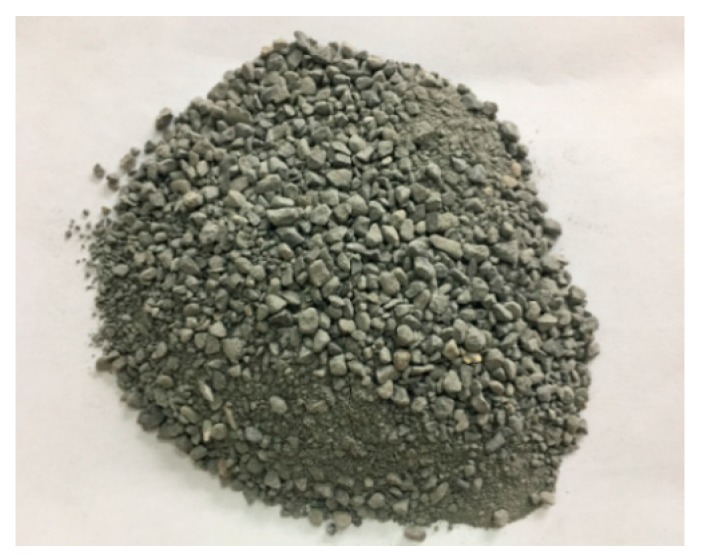
Manufactured sand.

**Figure 2 materials-12-01845-f002:**
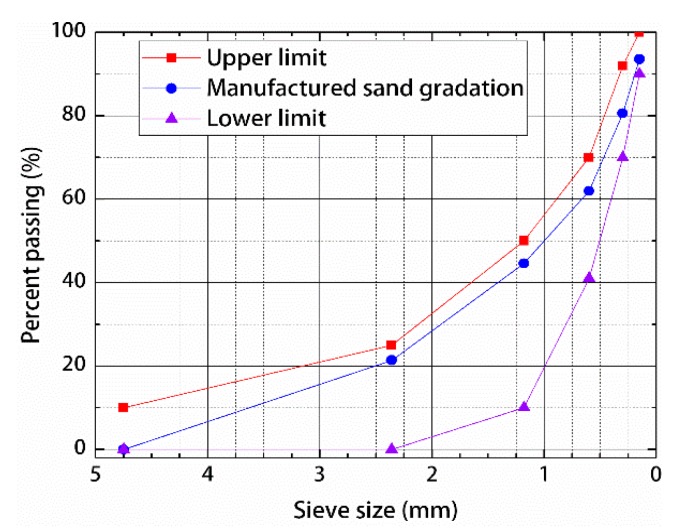
Gradation of manufactured sand used in this study.

**Figure 3 materials-12-01845-f003:**
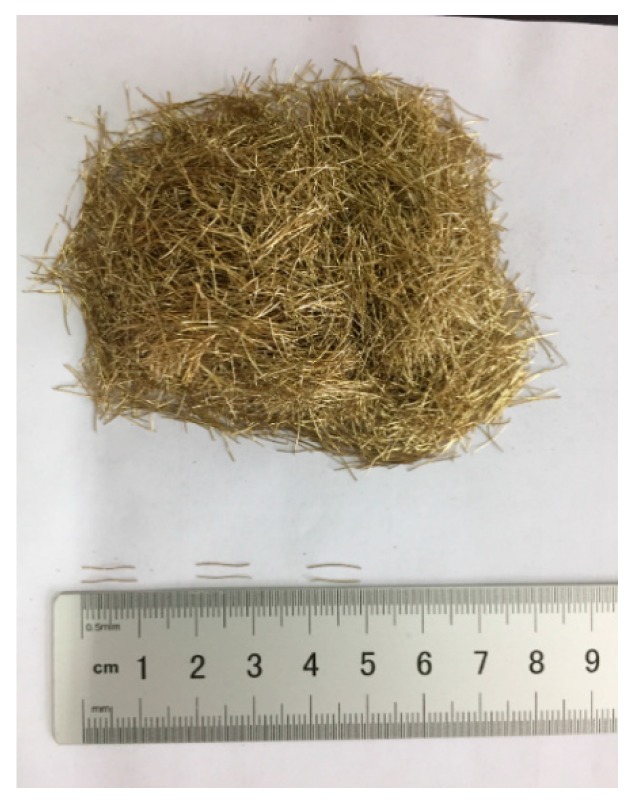
Steel fibers.

**Figure 4 materials-12-01845-f004:**
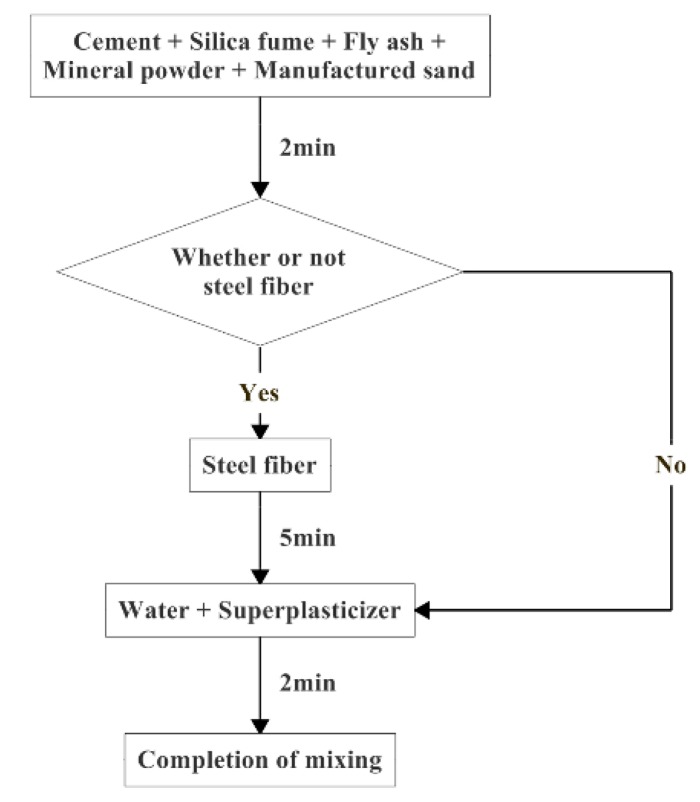
Preparation process.

**Figure 5 materials-12-01845-f005:**
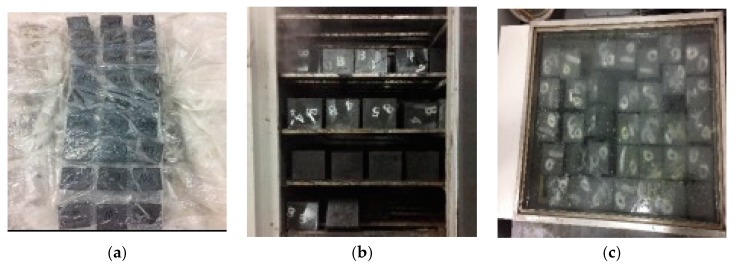
Curing methods: (**a**) natural curing, (**b**) standard curing, and (**c**) hot water curing.

**Figure 6 materials-12-01845-f006:**
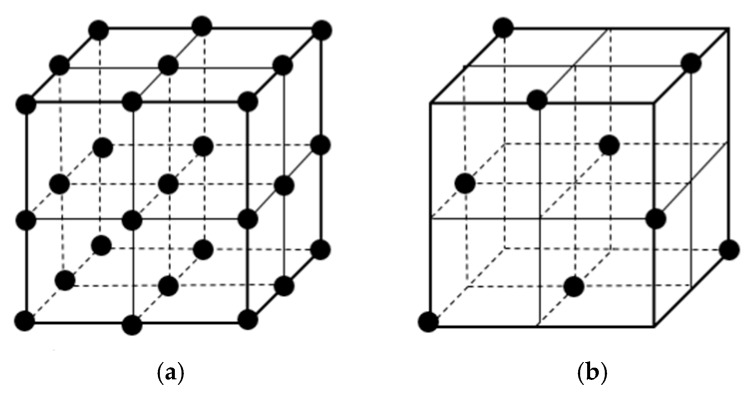
A four-factor layout: (**a**) comprehensive test; (**b**) orthogonal experimental design.

**Figure 7 materials-12-01845-f007:**
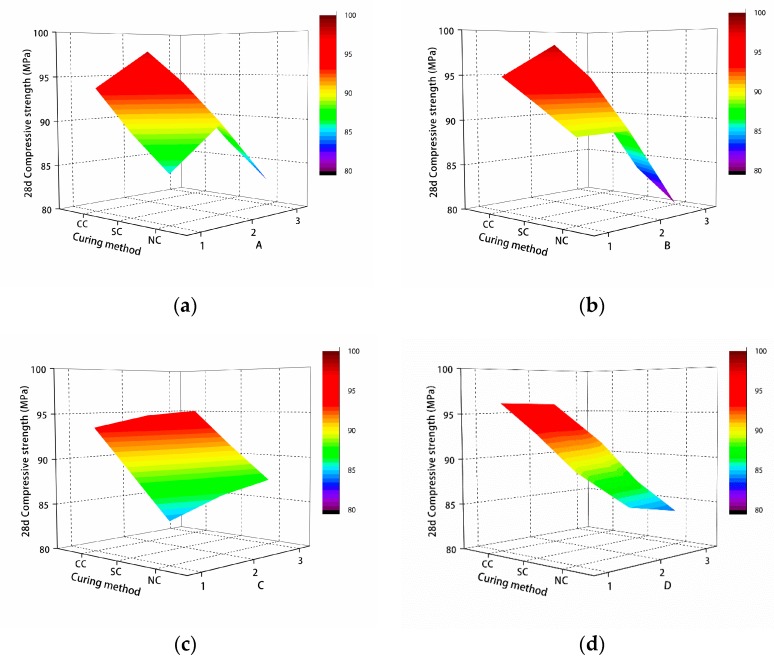
The relationship between various factors and compressive strength: (**a**) water/binder ratio, (**b**) silica fume content, (**c**) fly ash content, (**d**) and sand/binder ratio.

**Figure 8 materials-12-01845-f008:**
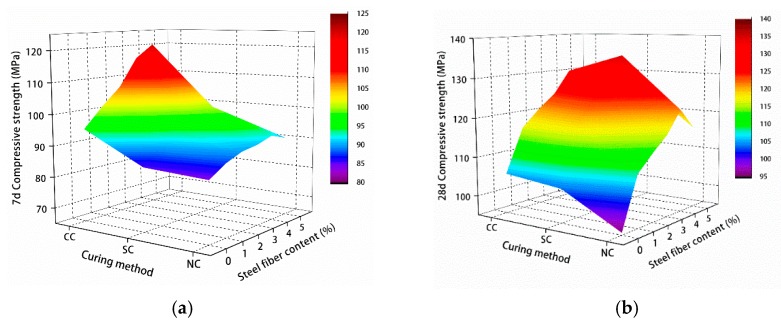
Compressive strength test results: (**a**) 7 and (**b**) 28 days.

**Figure 9 materials-12-01845-f009:**
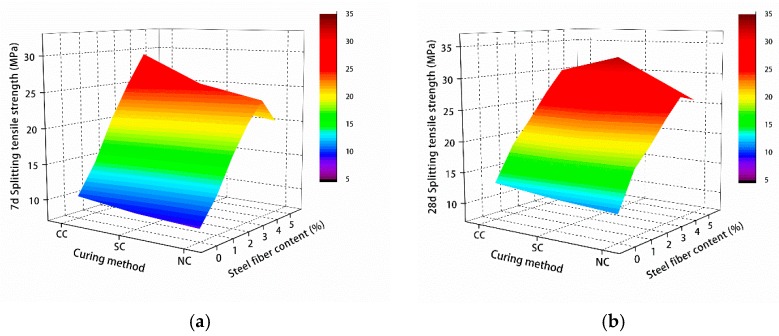
Splitting tensile strength test results: (**a**) 7 and (**b**) 28 days.

**Figure 10 materials-12-01845-f010:**
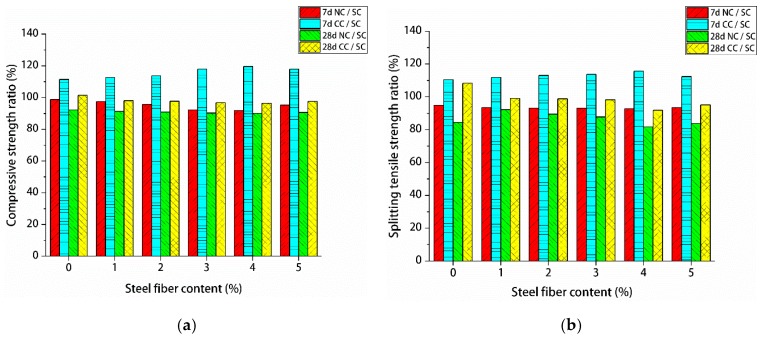
Manufactured sand reactive powder concrete (RPC) strength ratio: (**a**) manufactured sand RPC compressive strength ratio; (**b**) manufactured sand RPC splitting tensile strength ratio.

**Figure 11 materials-12-01845-f011:**
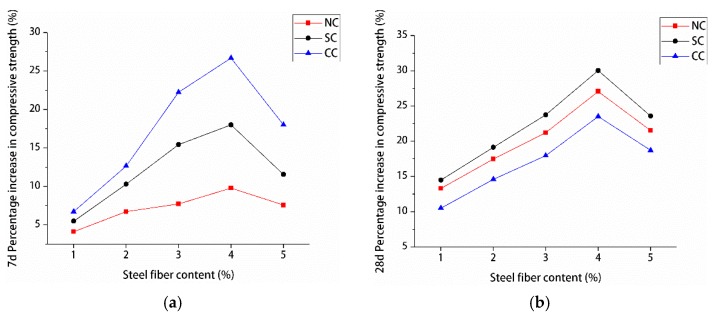
Compressive strength increase percentage: (**a**) 7 and (**b**) 28 days.

**Figure 12 materials-12-01845-f012:**
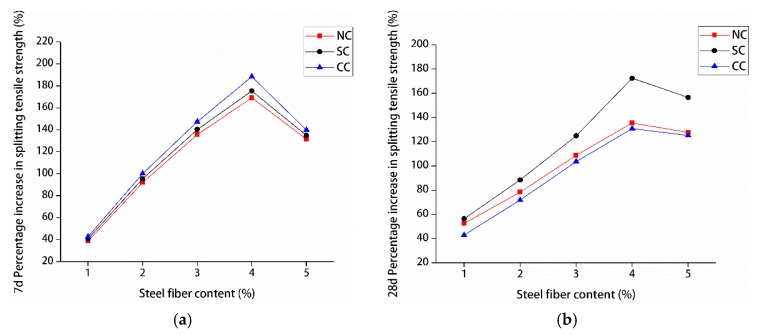
Splitting tensile strength increase percentage: (**a**) 7 and (**b**) 28 days.

**Figure 13 materials-12-01845-f013:**
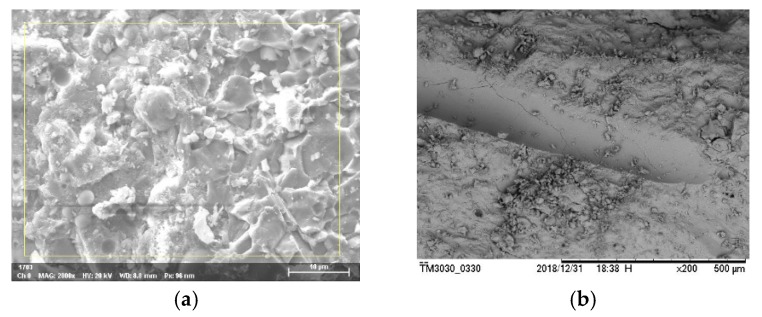
Manufactured sand RPC micropicture: (**a**) manufactured sand RPC matrix; (**b**) interface transition zone between steel fibers and the matrix.

**Figure 14 materials-12-01845-f014:**
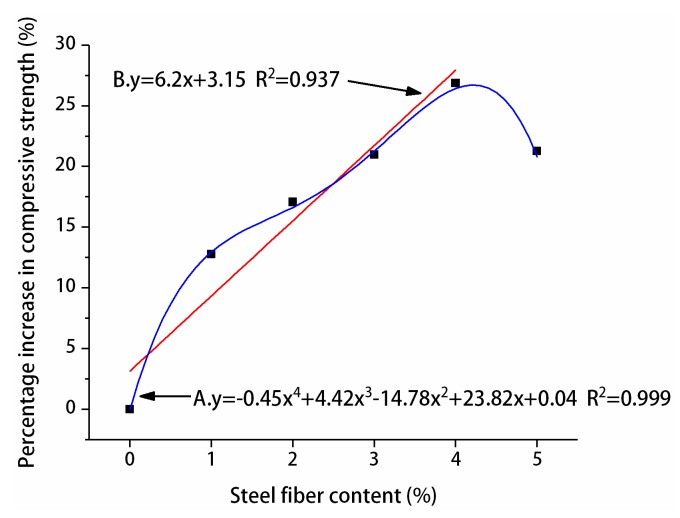
Relationship between compressive strength and steel fiber content.

**Figure 15 materials-12-01845-f015:**
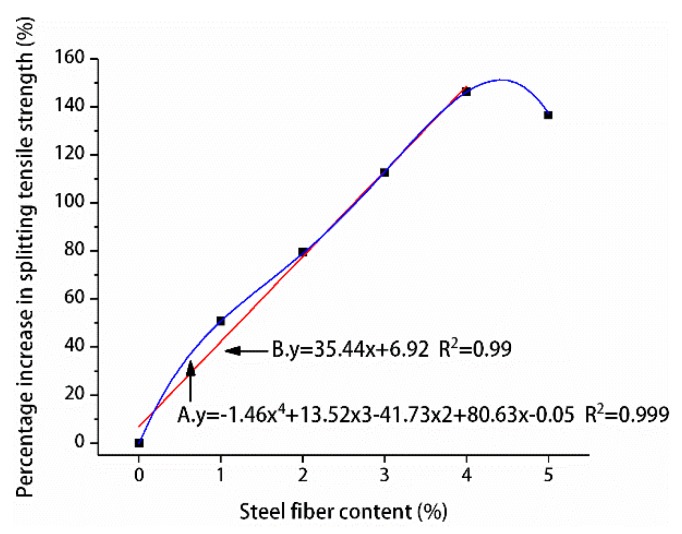
Relationship between splitting tensile strength and steel fiber content.

**Table 1 materials-12-01845-t001:** Physical and chemical properties of cement.

Properties		Standard Value	Actual Value
Physical properties	Specific surface area (m^2^/kg)	≥300	367
	Initial set (min.)	≥45	99
	Final set (min.)	≤390	145
Compressive strength	3 day (MPa)	≥23.0	29.1
	7 day (MPa)	≥52.5	
Flexural strength	3 day (MPa)	≥4.0	6.0
	7 day (MPa)	≥7.0	
Chemical properties	Stability	Qualified	Qualified
	Loss on ignition (%)	≤3.5	1.61
	MgO (%)	≤5.0	0.98
	SO_3_ (%)	≤3.5	2.62
	Insolubles (%)	≤1.5	1.01
	Cl^−^ (%)	≤0.06	0.007

**Table 2 materials-12-01845-t002:** Physical and chemical properties of silica fume.

Properties		Standard Value	Actual Value
Physical properties	Specific surface area (m^2^/kg)	≥15	20
	Pozzolanic activity index (%)	≥85	116
Chemical properties	SiO_2_ (%)	≥85.0	94.5
	Loss on ignition (%)	≤6.0	2.5
	Cl^−^ (%)	≤0.02	0.02
	Moisture content (%)	≤3.0	1.2
	Water demand ratio (%)	≤125	118

**Table 3 materials-12-01845-t003:** Chemical properties of fly ash.

Properties		Standard Value	Actual Value
Chemical properties	Water demand ratio (%)	≤105	94
	Loss on ignition (%)	≤8.0	1.0
	Moisture content (%)	≤1.0	0.1
	SO_3_ (%)	≤3.0	0.3
	CaO_3_ (%)	≤1.0	0.16
	MgO (%)	≤5.0	1.08
	Cl^−^ (%)	≤0.02	0.01

**Table 4 materials-12-01845-t004:** Physical and chemical properties of mineral powder.

Properties			Standard Value	Actual Value
Physical properties	Specific surface area(m^2^/kg)		≥400	429
	Liquidity ratio (%)		≥95	102
	Density (%)		≥2.8	2.9
Chemical properties	Moisture content (%)		≤1.0	0.1
	Loss on ignition (%)		≤3.0	1.07
	Pozzolanic activity index (%)	7 day	≥75	83
		28 day	≥95	98

**Table 5 materials-12-01845-t005:** Physical and chemical properties of superplasticizer.

Properties		Standard Value	Actual Value
Physical properties	Water reduction rate	≥25	28.5
	Gas content (%)	≤6.0	3.2
	Bleeding rate (%)	≤60	50
Chemical properties	Cl^−^ (%)	≤0.1	0.05
	OH^−^ (%)	≤3	1.2
	Na_2_SO_4_	≤0.5	0.06

**Table 6 materials-12-01845-t006:** Manufactured sand screening results.

Properties	Sieve size (mm)					
	2.36	1.18	0.6	0.3	0.15	Sieve bottom
Sieve residue (g)	107	116	87	93	65	32
Submeter (%)	21.4	23.2	17.4	18.6	13	6.4
Cumulative (%)	21.4	44.6	62	80.6	93.6	100

**Table 7 materials-12-01845-t007:** Physical properties of steel fiber.

Index	Diameter/mm	Length/mm	Aspect Ratio	Tensile Strength/MPa
Unit value	0.2	10	50	2850

**Table 8 materials-12-01845-t008:** Curing methods.

Curing Method		Specific Process
NC	Natural curing	Indoor natural curing 1-day demolding and maintained to 7 and 28 days under indoor natural curing conditions
SC	Standard curing	Standard curing 1-day demolding and maintained to 7 and 28 days under standard curing conditions
CC	Compound curing	Indoor natural curing 1-day demolding, hot water (60 ± 1 °C) for 1 day, and maintained to 7 and 28 days under indoor natural curing conditions

**Table 9 materials-12-01845-t009:** Factor level table.

Factors		Units	Levels		
			1	2	3
A	Water/binder ratio	–	0.16	0.18	0.2
B	Silica fume	%	10	13	16
C	Fly ash	%	10	15	20
D	Sand/binder ratio	–	0.7	0.75	0.8

**Table 10 materials-12-01845-t010:** Design and result of orthogonal experiment.

No.	Preparation Parameters				28-Day Compressive Strength (MPa)		
	Water/Binder Ratio	Silica Fume (%)	Fly Ash (%)	Sand/Binder Ratio	NC	SC	CC
1	0.16	10	10	0.7	89.98	92.76	95.01
2	0.16	13	15	0.75	87.54	94.36	99.49
3	0.16	16	20	0.8	78.4	80.35	86.76
4	0.18	10	15	0.8	90.92	92.98	95.21
5	0.18	13	20	0.7	97.55	102.65	104.13
6	0.18	16	10	0.75	81.4	87.3	94.1
7	0.2	10	20	0.75	86.08	90.2	94.33
8	0.2	13	10	0.8	82.06	87	91.47
9	0.2	16	15	0.7	80.56	83.49	89.33

**Table 11 materials-12-01845-t011:** Compressive strength range analysis table.

Curing Method	Factors	1	2	3	R
NC	A	85.31	89.96	82.90	7.06
	B	88.99	89.05	80.12	8.93
	C	84.48	86.34	87.34	2.86
	D	89.36	85.01	83.79	5.57
SC	A	89.16	94.31	86.90	7.41
	B	91.98	94.67	83.71	10.96
	C	89.02	90.28	91.07	2.05
	D	92.97	90.62	86.78	6.19
CC	A	93.75	97.81	91.71	6.10
	B	94.85	98.36	90.06	8.30
	C	93.53	94.68	95.07	1.55
	D	96.16	95.97	91.15	5.01

**Table 12 materials-12-01845-t012:** Test results.

Age	Curing Method	Steel Fiber Content (%)					
		0	1	2	3	4	5
7-day compressive strength (MPa)	NC	84.18	87.64	89.84	90.69	92.42	90.55
	SC	85.19	89.87	93.96	98.34	100.52	95.03
	CC	94.93	101.31	106.95	116.04	120.27	112.03
28-day compressive strength (MPa)	NC	95.60	108.32	112.31	115.89	121.48	116.17
	SC	103.68	118.70	123.53	128.30	134.83	128.13
	CC	105.28	116.35	120.65	124.19	130.05	124.98
7-day splitting tensile strength (MPa)	NC	8.75	12.15	16.81	20.63	23.56	20.27
	SC	9.22	12.99	18.03	22.16	25.39	21.66
	CC	10.18	14.54	20.39	25.19	29.37	24.35
28-day splitting tensile strength (MPa)	NC	11.35	17.32	20.25	23.71	26.74	25.85
	SC	12.01	18.79	22.64	27.01	32.72	30.82
	CC	13.02	18.61	22.38	26.52	30.06	29.32

**Table 13 materials-12-01845-t013:** Calculated values of compressive strength of manufactured sand RPC at 28 days.

Steel Fiber Content (%)	$\lambda_{t}\;(\%)$	NC (MPa)	Error (%)	SC (MPa)	Error (%)	CC (MPa)	Error (%)
0	3.15	98.61	3.15	106.95	3.15	108.60	3.15
1	9.35	104.54	−3.49	113.37	−4.49	115.12	−1.05
2	15.55	110.47	−1.64	119.80	−3.02	121.65	0.83
3	21.75	116.39	0.43	126.23	−1.61	128.18	3.21
4	27.95	122.32	0.69	132.66	−1.61	134.71	3.58

**Table 14 materials-12-01845-t014:** RPC compressive strength [28].

Curing Method		Steel Fiber Content (%)			
		0	1	2	3.5
Natural curing	Actual value (MPa)	93.5	102.1	115.8	112.3
	Calculated value (MPa)	96.45	102.24	108.04	116.73
	Error (%)	3.15	0.14	−6.70	3.95
Standard curing	Actual value (MPa)	101.7	113.4	125.2	133.6
	Calculated value (MPa)	104.90	111.21	117.51	126.97
	Error (%)	3.15	−1.93	−6.14	−4.96
Hot water curing	Actual value (MPa)	108.60	115.12	121.65	128.18
	Calculated value (MPa)	113.4	121.2	132.7	144.3
	Error (%)	3.15	2.31	−1.26	−1.89
Steam curing	Actual value (MPa)	124.9	135.6	143.2	160.8
	Calculated value (MPa)	128.83	136.58	144.32	155.94
	Error (%)	3.15	0.72	0.78	-3.02

**Table 15 materials-12-01845-t015:** RPC compressive strength [17].

Steel Fiber Content (%)	Standard Curing		
	Actual Value (MPa)	Calculated Value (MPa)	Error (%)
0	88.49	91.28	3.15
0.5	105.37	94.02	−10.77
1.5	112.29	99.51	−11.38
2.5	128.91	104.99	−18.55
3.5	132.32	110.48	−16.51

**Table 16 materials-12-01845-t016:** Calculated values of splitting tensile strength of manufactured sand RPC at 28 days.

Steel Fiber Content (%)	$\lambda_{t}\;(\%)$	NC (MPa)	Error (%)	SC (MPa)	Error (%)	CC (MPa)	Error (%)
0	6.92	12.14	6.92	12.84	6.92	13.92	6.92
1	42.36	16.16	−6.71	17.10	−9.01	18.54	−0.40
2	77.80	20.18	−0.34	21.35	−5.68	23.15	3.44
3	113.24	24.20	2.08	25.61	−5.18	27.76	4.69
4	148.68	28.23	5.55	29.87	−8.72	32.38	7.71

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
