# Peer review of "Study on Mix Proportion Optimization of Manufactured Sand RPC and Design Method of Steel Fiber Content under Different Curing Methods"

_materials, 2019, doi:10.3390/ma12111845_

Round 1
Reviewer 1 Report
The paper investigates the influence of adding a certain type of steel fibers to reactive powder concrete (RPC) containing manufactured sand.
The paper discusses only the influence of the amount of fibers on the compressive and tensile splitting strength of RPC, fiber orientations are not discussed. Figure 12c not only shows clustering of fibers, but also seems to show a preference direction.
References:
- spacing next to punctuation is often missing
- list seems very focussed on Asian authors, it would be good to cite also European authors. The Asian and European research communities sometimes seem to be unaware of each other's work (at least the impression arises when looking at the citations), it would be beneficial to also cite authors from the other community.
Ref 6: incomplete
Author Response
Response to Reviewer 1 Comments
Thank you for your advice. All your suggestions are very important and have important guiding significance for my thesis writing and scientific research work. I have revised the manuscript on the basis of your suggestions and carefully proofed the manuscript. The following is a description of the changes based on your comments:
Point 1: The paper discusses only the influence of the amount of fibers on the compressive and tensile splitting strength of RPC, fiber orientations are not discussed. Figure 12c not only shows clustering of fibers, but also seems to show a preference direction
Response 1: Since the stirring process in this study could not control the fiber orientation, the fiber orientation was not analyzed, so I will delete figure 12(c).
Point 2: References:
- spacing next to punctuation is often missing
- list seems very focussed on Asian authors, it would be good to cite also European authors. The Asian and European research communities sometimes seem to be unaware of each other's work (at least the impression arises when looking at the citations), it would be beneficial to also cite authors from the other community.
Ref 6: incomplete
Response 2: The references have been modified according to your Suggestions, as follows:
References
1. Richard, P.; Cheyrezy, M.H. Composition of reactive powder concretes. J. Cement and Concrete Research. 1995, 25, 1501-1511.
2. Dugat, J.; Roux ,N.; Bernier, G. Mechanical properties of reactive powder concretes. J. Materials and structures, 1996, 29, 233-240.
3. Cheyrezy, M.; Maret, V.; Frouin, L. Microstructural analysis of RPC(reactive powder concrete). J. Cement and concrete Research. 1995, 25, 1491-1500.
4. Chen, M.; Zheng, W. A Study on Optimum Mixture Ratio of Reactive Powder Concrete. J. Advances in Materials Science and Engineering, 2018, 2018.
5. Pang, J.C.; Liu, R.G. Improvement of performance of ultra-high performance concrete based composite material added with nano materials. J. Frattura ed Integrità Strutturale, 2016, 10, 130-138.
6. Hiremath, P.N.; Yaragal, S.C. Investigation on Mechanical Properties of Reactive Powder Concrete under Different Curing Regimes. J. Materials Today: Proceedings, 2017, 4, 9758-9762.
7. Yao, Y.F.; Guo X.Y.; W.P.; Peng X.Y. Study on the Influence of Particle Gradation on the Performance of River Sand RPC. J. Concrete World, 2017, 72-76. (In Chinese)
8. Su, Y.; Wu, C.; Li, J. Development of novel ultra-high performance concrete: From material to structure. J. Construction and Building Materials, 2017, 135, 517-528.
9. Wu, Z.; Shi, C.; He, W. Effects of steel fiber content and shape on mechanical properties of ultra high performance concrete. J. Construction and building materials, 2016, 103, 8-14.
10. Qi, J.C. Experiment on strength and pore deterioration of high performance concrete with manufactured sand after high temperature. J. Chinese science and technology paper, 2018, 13, 2219-2222. (In Chinese)
11. Bai, Y.Z.; Wan, X.M.; Zhao, T.J.; Li, H.; Gao, S.; Yin, G.L. Preparation technology of self-compacting concrete for high-content sand low-gelling materials. J. Construction Technology, 2016, 45, 30-33. (In Chinese)
12. Liu, G.B.; Yu, Q.; Liu, Z.X.; Li, H.R. Experimental Study on Preparation of Lightweight Aggregate Concrete by Mechanized Sand. J. Journal of Qingdao Technological University, 2019, 40, 138-144. (In Chinese)
13. Hiremath, P.N.; Yaragal, S.C. Effect of different curing regimes and durations on early strength development of reactive powder concrete. J. Construction and Building Materials, 2017, 154, 72-87.
14. Helmi, M.; Hall, M.R.; Stevens, L.A. Effects of high-pressure/temperature curing on reactive powder concrete microstructure formation. J. Construction and Building Materials, 2016, 105, 554-562.
15. Tam, C.; Tam, V.W. Microstructural behaviour of reactive powder concrete under different heating regimes. J. Magazine of concrete research, 2012, 64, 259-267.
16. Ipek, M.; Yilmaz, K.; Sümer, M. Effect of pre-setting pressure applied to mechanical behaviours of reactive powder concrete during setting phase. J. Construction and Building Materials, 2011, 25, 61-68.
17. MA, K.Z.; Que, A.; Liu, C. Analysis of the Influence of Steel Fiber Content on Mechanical Properties of Reactive Powder Concrete. J. Concrete, 2016, 76-79+83. (In Chinese)
18. Ren, G.M.; Wu, H.; Fang, Q. Effects of steel fiber content and type on static mechanical properties of UHPCC. J. Construction and Building Materials, 2018, 163, 826-839.
19. Jiang, Y.; Chen, T.T.; Du, H.X. Effect of steel fiber on mechanical properties of reactive powder concrete. J. Chinese journal of silicate, 2017, 36, 2173-2177. (In Chinese)
20. Guo, J.X. Effects of different fibers on compressive strength of reactive powder concrete. J. Concrete, 2016, 87-90. (In Chinese)
21. Ramyashree, P.; Subathra, S. Experimental study on developing ultra high strength concrete using reactive powder concrete. J.
22. Ju, Y.Z.; Wang, D.H.; Li, Q.C.; Jia, Y.Z.; Xiao, Q. Effect of steel fiber content on mechanical properties of reactive powder concrete. J. Experimental mechanics, 2011, 26, 254-260. (In Chinese)
23. Wu, Z.; Shi, C.; He, W. Effects of steel fiber content and shape on mechanical properties of ultra high performance concrete. J. Construction and building materials, 2016, 103, 8-14.
24. Huang, H.; Gao, X.; Li, L. Improvement effect of steel fiber orientation control on mechanical performance of UHPC. J. Construction and Building Materials, 2018, 188, 709-721.
25. GB175-2007. Standard for Common Portland Cement; Chinese Standard: Beijing, China, 2007. (In Chinese)
26. GB/T51003/2014. Technical code for application of mineral admixture; Chinese Standard: Beijing, China, 2014. (In Chinese)
27. JGJ 52-2006. Standard for Technical Requirements and Test Method of Sand and Rrushed Stone (or gravel) for Ordinary Concrete; Ministry of Construction of the People's Republic of China: Beijing, China, 2015. (In Chinese)
28. GB/T31387-2015. Standard for Reactive Powder Concrete; Chinese Standard: Beijing, China, 2015. (In Chinese)
29. Guo, T.X.; Teng, T.J.; Yu, Q.K. Influence of steel fiber content on RPC strength and toughness under different curing conditions. J. Urban housing, 2016, 23, 119-121. (In Chinese)
Thank you again for your advice and hope to learn more from you.

Reviewer 2 Report
In this paper the authors use manufactured sand as fine aggregate to prepare RPC and they investigates the mechanical properties of samples and the design method of steel fiber content. Compressive and splitting tensile strengths are evaluated under different conditions through laboratory tests and scanning electron microscopy (SEM).
The paper is interesting and well written. Only some minor points should be addressed:
1) Fig. 2 should be replaced with a figure of better resolution. The fibers showed are quite blurred.
2) the bibliography should be improved with other studies on the same topic. See for example:
- Ren, G. M., Wu, H., Fang, Q., & Liu, J. Z. (2018). Effects of steel fiber content and type on static mechanical properties of UHPCC. Construction and Building Materials, 163, 826-839.
- Huang, H., Gao, X., Li, L., & Wang, H. (2018). Improvement effect of steel fiber orientation control on mechanical performance of UHPC. Construction and Building Materials, 188, 709-721.
Author Response
Response to Reviewer 2 Comments
Thank you for your advice. All your suggestions are very important and have important guiding significance for my thesis writing and scientific research work. I have revised the manuscript on the basis of your suggestion and carefully proofed the manuscript. The following is a description of the changes based on your comments:
Point 1: Fig. 2 should be replaced with a figure of better resolution. The fibers showed are quite blurred.
Response 1: The fiber picture has been retaken and modified as follows:
Point 2: the bibliography should be improved with other studies on the same topic. See for example:
- Ren, G. M., Wu, H., Fang, Q., & Liu, J. Z. (2018). Effects of steel fiber content and type on static mechanical properties of UHPCC. Construction and Building Materials, 163, 826-839.
- Huang, H., Gao, X., Li, L., & Wang, H. (2018). Improvement effect of steel fiber orientation control on mechanical performance of UHPC. Construction and Building Materials, 188, 709-721
Response 2: Other studies on the same topic have been referenced according to your suggestion, as follows:
[23] Wu, Z.; Shi, C.; He, W. Effects of steel fiber content and shape on mechanical properties of ultra high performance concrete. J. Construction and building materials, 2016, 103, 8-14.
[24] Huang, H.; Gao, X.; Li, L. Improvement effect of steel fiber orientation control on mechanical performance of UHPC. J. Construction and Building Materials, 2018, 188, 709-721.
Thank you again for your advice and hope to learn more from you.

Reviewer 3 Report
Generally, this is an interesting Manufactured Sand Reactive Powder (MSRP) concrete with a steel fiber for the prediction of Compressive strength and splitting tensile strength. This study seems reasonable. However, the following were apparent:
1. The title shows the effect of the mechanical properties of the concrete based on MSRP, but the manuscript explained the effect of the concrete properties of the concrete with the curing methods such as natural, standard, and compound curing.
2. Line 86 and 91 should add references for the material properties.
3. Line 106, should add the reference on figure 1.
4. Line 113, the author explains the fiber contents but should make it clear whether the amount of fiber is based on either volume or mass.
5. Line 116, in table 8, not enough data sets. All variables have been fixed but the amount of fiber, 0%, 1, 2, 3, 4, and 5%. If the author wants to find how to affect the MSRP on the concrete hardened properties, redesign the mix ratio, different amount of MSRP percentages should be added as well.
6. Line 128, should add the reference.
7. Line 152 and 166, in figures 5 and 6, unable to recognize the failure mode with the pictures. I would recommend that you should delete the figures.
8. Line 183, in Figure 9, can you explain the meaning of 100% and whether if it’s a normalized number?
9. In figure 10 and 11, should be the same pattern and data with figure 7 and 8. Recommend delete figure 10 and 11.
10. In chapter 4, That is a critical statement considering the very few samples with the fiber content up to 4% that were used to model the prediction formula that only used w/b ratio of 0.18. It would be best if the authors provided a better explanation of their methodology, express the limitations of this study, and/or even rethink their approach in the making of this prediction model.
11. Nothing new on using a single linear regression equation for strength prediction.
12. The conclusions are predictable, not very impressive, and too short.
Author Response
Response to Reviewer 3 Comments
Thank you for your advice. All your suggestions are very important and have important guiding significance for my thesis writing and scientific research work. I have revised the manuscript on the basis of your suggestion and carefully proofed the manuscript. The following is a description of the changes based on your comments:
Note: I will replace the three curing methods in this paper with abbreviations, natural curing (NC), standard curing (SC), composite curing (CC)
Point 1: The title shows the effect of the mechanical properties of the concrete based on MSRP, but the manuscript explained the effect of the concrete properties of the concrete with the curing methods such as natural, standard, and compound curing.

Response 1: According to your Suggestions, the title of this paper has been modified as follows:
Study on Mix Proportions Optimization of Manufactured Sand RPC and Design Method of Steel Fiber Content under Different Curing Methods
Point 2: Line 86 and 91 should add references for the material properties.
Response 2: References to material properties have been added as recommended, as follows:
the general Portland Cement Inspection Standard (GB175-2007) [25]
the Application of Mineral Admixtures (GB/T51003/2014) [26]
[25]GB175-2007. Standard for Common Portland Cement; Chinese Standard: Beijing, China, 2007. (In Chinese)
[26]GB/T51003/2014. Technical code for application of mineral admixture; Chinese Standard: Beijing, China, 2014. (In Chinese)
Point 3: Line 106, should add the reference on figure 1
Response 3: References have been added as suggested by you, as follows:
the Standard for Technical Requirements and Test Method of Sand and Crushed Stone for Ordinary Concrete (JGJ 52-2006) [27]
[27]JGJ 52-2006. Standard for Technical Requirements and Test Method of Sand and Rrushed Stone (or gravel) for Ordinary Concrete; Ministry of Construction of the People's Republic of China: Beijing, China, 2015. (In Chinese)
Point 4: Line 113, the author explains the fiber contents but should make it clear whether the amount of fiber is based on either volume or mass.
Response 4: The steel fiber content in this study is volume content, which has been modified in this paper. The modified content is as follows:
On the basis of the optimal matrix mix proportions of manufactured sand RPC, steel fibers with volume content of 0%, 1%, 2%, 3%, 4% and 5% were added.
Point 5: Line 116, in table 8, not enough data sets. All variables have been fixed but the amount of fiber, 0%, 1, 2, 3, 4, and 5%. If the author wants to find how to affect the MSRP on the concrete hardened properties, redesign the mix ratio, different amount of MSRP percentages should be added as well.
Response 5: This study was based on the experiment of determining the matrix proportion of the manufactured sand RPC matrix, and then the steel fiber with different content was added for research. Since the author focused on the study of the steel fiber content in the manuscript last time, the test results and analysis of determining the matrix proportion were not written in the paper. According to your suggestion, I will add the test results and analysis of the manufactured RPC matrix mix proportion to the text. The contents are as follows:
This study designed a four-factor and three-level orthogonal test. As shown in Table 9, Water/binder ratio, silica fume content, fly ash content and sand/binder ratio were taken as four factors, and three levels were set for each factor.
Table 9. Factor level table.
Factors | Units | Levels | |||
1 | 2 | 3 | |||
A | Water/binder ratio | – | 0.16 | 0.18 | 0.2 |
B | Silica fume | % | 10 | 13 | 16 |
C | Fly ash | % | 10 | 15 | 20 |
D | Sand/binder ratio | – | 0.7 | 0.75 | 0.8 |
The effects of water/binder ratio, silica fume content, fly ash content and sand/binder ratio on the 28 d compressive strength of manufactured sand RPC matrix were studied by OED. The experimental design and results are shown in Table 10.
Table 10. Design and result of orthogonal experiment
No. | Preparation Parameters | 28 d Compressive strength (MPa) | |||||
Water/binder ratio | Silica fume (%) | Fly ash (%) | Sand/binder ratio | NC | SC | CC | |
1 | 0.16 | 10 | 10 | 0.7 | 89.98 | 92.76 | 95.01 |
2 | 0.16 | 13 | 15 | 0.75 | 87.54 | 94.36 | 99.49 |
3 | 0.16 | 16 | 20 | 0.8 | 78.4 | 80.35 | 86.76 |
4 | 0.18 | 10 | 15 | 0.8 | 90.92 | 92.98 | 95.21 |
5 | 0.18 | 13 | 20 | 0.7 | 97.55 | 102.65 | 104.13 |
6 | 0.18 | 16 | 10 | 0.75 | 81.4 | 87.3 | 94.1 |
7 | 0.2 | 10 | 20 | 0.75 | 86.08 | 90.2 | 94.33 |
8 | 0.2 | 13 | 10 | 0.8 | 82.06 | 87 | 91.47 |
9 | 0.2 | 16 | 15 | 0.7 | 80.56 | 83.49 | 89.33 |
The test results were analyzed by the range analysis method, and the range effect value of water/binder ratio, silica fume content, fly ash content and sand/binder ratio on the 28d compressive strength of the manufactured sand RPC matrix were analyzed. The results are shown in Table 11.
Table 11. Compressive strength range analysis table
Curing method | Factors | 1 | 2 | 3 | R |
NC | A | 85.31 | 89.96 | 82.90 | 7.06 |
B | 88.99 | 89.05 | 80.12 | 8.93 | |
C | 84.48 | 86.34 | 87.34 | 2.86 | |
D | 89.36 | 85.01 | 83.79 | 5.57 | |
SC | A | 89.16 | 94.31 | 86.90 | 7.41 |
B | 91.98 | 94.67 | 83.71 | 10.96 | |
C | 89.02 | 90.28 | 91.07 | 2.05 | |
D | 92.97 | 90.62 | 86.78 | 6.19 | |
CC | A | 93.75 | 97.81 | 91.71 | 6.10 |
B | 94.85 | 98.36 | 90.06 | 8.30 | |
C | 93.53 | 94.68 | 95.07 | 1.55 | |
D | 96.16 | 95.97 | 91.15 | 5.01 |
The value of the range value R indicates the influence of this factor on the test index. It can be seen from the range analysis of compressive strength in table 12 that the primary and secondary relationships of the influence of four factors on the 28d compressive strength of the RPC matrix under NC, SC and CC were all B > A > D > C–i.e., silica fume content > water/binder ratio > sand/binder ratio > fly ash content. Mixing proportions of manufactured sand RPC matrix is A2B2C3D1–i.e., water/binder ratio: 0.18, silica fume content: 13%, fly ash content: 20% and sand/binder ratio: 0.7.
Point 6: Line 128, should add the reference.
Response 6: References have been added as suggested by you, as follows:
GB/T31387-2015 [28]
[28]GB/T31387-2015. Standard for Reactive Powder Concrete; Chinese Standard: Beijing, China, 2015. (In Chinese)
Point 7: Line 152 and 166, in figures 5 and 6, unable to recognize the failure mode with the pictures. I would recommend that you should delete the figures.
Response 7: The picture has been deleted according to your suggestion
Point 8: Line 183, in Figure 9, can you explain the meaning of 100% and whether if it’s a normalized number?
Response 8: 100% is based on standard curing, and the ratio of strength under natural curing and compound curing to that under standard curing. The picture of the manuscript was not clearly expressed last time, so the author modified the picture as follows:
| |
(a) | (b) |
Point 9: In figure 10 and 11, should be the same pattern and data with figure 7 and 8. Recommend delete figure 10 and 11.
Response 9: The picture in the manuscript was not clearly expressed last time. The author modified the picture as follows:
(a) | (b) |
Figure 10. Compressive strength increase percentage: (a)7 d; (b) 28 d.
(a) | (b) |
Point 10: In chapter 4, That is a critical statement considering the very few samples with the fiber content up to 4% that were used to model the prediction formula that only used w/b ratio of 0.18. It would be best if the authors provided a better explanation of their methodology, express the limitations of this study, and/or even rethink their approach in the making of this prediction model.
Response 10:
Point 11: Nothing new on using a single linear regression equation for strength prediction.
Response 11: The author added curve fitting and compared it with linear fitting. The author thinks that linear fitting is easy to be generalized and modified as follows:
When the content of steel fiber is 5%, the preparation cost increases and the compressive strength decreases. It is suggested that the content of steel fiber should be between 0% and 4% in practical engineering. Because the contribution rate of steel fiber content 0%–4% steel fiber content to the compressive strength of manufactured sand RPC is close to the straight line, in order to facilitate the engineering design, linear fitting is adopted in this paper, as shown in A of Figure 13.
Figure 13. Figure 14.
Point 12: The conclusions are predictable, not very impressive, and too short.
Response 12: The author has enriched and refined the conclusions and revised them as follows:
4 Conclusion
In this study, manufactured sand is used as fine aggregate to prepare RPC under natural, standard, and compound curing conditions. The mechanical properties of manufactured sand and design of steel fiber content are investigated through the evaluation of compressive and split tensile strengths. The conclusions are summarized as follows.
· Based on the orthogonal experimental design, the optimum mixing proportions of manufactured sand RPC matrix is water/binder ratio: 0.18, silica fume content: 13%, fly ash content: 20% and sand/binder ratio: 0.7.
· The compressive and splitting tensile strengths of manufactured sand RPC increase to a certain extent, the impact of the increase of compressive strength is small, and the increase in splitting tensile strength is significant when the steel fiber content is 0%-4%. SEM analysis shows that the porosity of manufactured sand RPC is extremely low, the hydration products are mostly C–S–H gel, the interface transition zone between steel fibers and the matrix is close, and agglomeration occurs when the steel fiber content is 5%.
· Compared with standard curing, compound curing improves the early strength of manufactured sand RPC and do not improve the later strength. Natural curing basically meets the engineering requirements and is beneficial to practical applications although its strength is slightly lower than standard curing. The percentage of compressive strength and splitting tensile strength increase of manufactured sand RPC 7 d under three curing methods is CC > SC > NC, and the order of increase of compressive strength and splitting tensile strength of 28 d is SC > NC > CC.
· Prediction formulas of 28 d compressive and splitting tensile strengths of steel fibers with 0%–4% are established to aid the selection of steel fiber content based on different actual demands.
Thank you again for your advice and hope to learn more from you.

Round 2
Reviewer 3 Report
Thank you for your quick response.
1. You have added more test variables such as Fly ash, etc. It appears to be fine. However, I could not find any explanations and comparisons of the test results under the section.
2. You have changed from single regression to poly nominal. When last we spoke, I had meant that the prediction model should include more variables such as aspect ratio, types of binder, W/B ratios, and types of curing.
Author Response
Response to Reviewer 3 Comments
Thank you for your advice. All your suggestions are very important and have important guiding significance for my thesis writing and scientific research work. I have revised the manuscript on the basis of your suggestion and carefully proofed the manuscript. The following is a description of the changes based on your comments:
Point 1:You have added more test variables such as Fly ash, etc. It appears to be fine. However, I could not find any explanations and comparisons of the test results under the section.
Response 1: It has been modified according to your suggestions, as follows:
| |
(a) | (b) |
(c) | (d) |
Figure 7. The relationship between various factors and compressive strength: (a) Water/binder ratio; (b) Silica fume; (c) Fly ash; (d) Sand/binder ratio. | |
As shown in Figure 7 (a), as the water/binder ratio decreased, the 28d compressive strength of the manufactured sand RPC matrix first increased and then decreased. This is because under the conditions of compaction, the lower the water/binder ratio, the higher the compressive strength of the matrix. However, when the water/binder ratio is 0.16, because the water/binder ratio is too low, the water is not enough to fully hydrate the cement, resulting in a decrease in the compressive strength of the matrix.
As shown in Figure 7 (b), with the increase of silica fume content, the 28d compressive strength of the manufactured sand RPC matrix first increased and then decreased. This is because of the micro-aggregate effect of silica fume. Compared with cement, silica fume is thinner and can be fully filled into the cement gap, which greatly promotes the density of cement matrix and increases its compressive strength.
As shown in Figure 7 (c), with the increase of fly ash content, the 28d compressive strength of the manufactured sand RPC matrix gradually increased. This is because the particle size of fly ash is small, and the surface is smooth and spherical. As the mixing goes on, fly ash produces "ball effect", which plays a role in filling the particle gap and improving the density of slurry, thus improving the compressive strength of the matrix.
As shown in Figure 7 (d), the 28d compressive strength of the manufactured sand RPC matrix gradually decreased with the increase of the sand/binder ratio. The effect of sand/binder ratio on the compressive strength of matrix is essentially the effect of average slurry thickness on the compressive strength of matrix. When the sand/binder ratio decreases, the average slurry thickness increases, so that there are enough slurry between the sand grains to connect them as a whole, and the bonding force between the sand grains and the slurry also increases, so that the compressive strength of the matrix increases. On the contrary, the compressive strength of matrix decreases with the increase of sand/binder ratio and the decrease of average slurry thickness.
Point 2: You have changed from single regression to poly nominal. When last we spoke, I had meant that the prediction model should include more variables such as aspect ratio, types of binder, W/B ratios, and types of curing.
Response 2:
, (1)
where f is 28 d compressive strength (MPa),is steel fiber compressive strength contribution, , f0 is corresponding to the 28 d compressive strength of undoped steel fiber under curing mode (MPa), x is steel fiber content (%).
, (2)
where f is 28 d splitting tensile strength (MPa), is steel fiber splitting tensile strength contribution, , f0 is corresponding to the 28 d splitting tensile strength of undoped steel fiber under curing mode (MPa), x is steel fiber content (%).
In this study, is the compressive strength or the splitting tensile strength of the matrix. Based on the determination of the compressive strength and the splitting tensile strength of the RPC matrix, the compressive strength and the splitting tensile strength that can be achieved with different steel fiber content are predicted. Based on the compressive strength and split tensile strength of RPC matrix, the compressive strength and split tensile strength of different steel fiber content under the three curing methods are unified. After the unification, it is found that different curing methods are suitable for this model. The experimental results of other researchers have been verified, and the errors are small. Therefore, I think this formula is of great significance.
Thank you again for your advice and hope to learn more from you.
